# Can Exercise Help Regulate Blood Pressure and Improve Functional Capacity of Older Women with Hypertension against the Deleterious Effects of Physical Inactivity?

**DOI:** 10.3390/ijerph18179117

**Published:** 2021-08-29

**Authors:** Luis Leitão, Moacir Marocolo, Hiago L. R. de Souza, Rhai André Arriel, João Guilherme Vieira, Mauro Mazini, Hugo Louro, Ana Pereira

**Affiliations:** 1Sciences and Technology Department, Superior School of Education of Polytechnic Institute of Setubal, 2910-761 Setúbal, Portugal; ana.fatima.pereira@ese.ips.pt; 2Life Quality Research Centre, 2040-413 Rio Maior, Portugal; 3Post Graduate Program in Physical Education, Federal University of Juiz de For a, Juiz de Fora 36036-900, Brazil; isamjf@gmail.com (M.M.); hlrsouza@gmail.com (H.L.R.d.S.); rhaiarriel@bol.com.br (R.A.A.); joaoguilhermevds@gmail.com (J.G.V.); 4Graduate Program in Physical Education—Sudamerica Faculty, Cataguases 36774-552, Brazil; personalmau@hotmail.com; 5Sports Science School of Rio Maior, Polytechnic Institute of Santarém, 2040-413 Rio Maior, Portugal; hlouro@esdrm.ipsantarem.pt

**Keywords:** older women, multicomponent exercise program, detraining, hypertension

## Abstract

Background: Sedentarism and inactivity are risk factors for the development of hypertension. Thus, the prevention of the natural process of biological and physiological aging of older women through physical exercise results in higher benefits in preventing cardiovascular diseases and can be a key factor for its treatment. Multicomponent exercise (METP) is a training method that may help older women with hypertension by improving their quality of life and their response to treatment. Methods: Twenty-eight older Caucasian women with hypertension (66.7 ± 5.3 years, 1.59 ± 0.11 m) performed a supervised METP program of nine months followed by three months of detraining (DT), and seventeen older women (68.2 ± 4.7 years, 1.57 ± 0.16 cm) with hypertension maintained their daily routine, without exercise. Blood pressure (BP), resting heart rate, and functional capacity (FC) were evaluated at the beginning and at the end of the program, and after three months of DT. Results: The ME program improved (*p* < 0.05) systolic BP (−5.37%), diastolic BP (−5.67%), resting heart rate (−7.8%), agility (9.8%), lower body strength (27.8%), upper body strength (10.0%), and cardiorespiratory capacity (8.6%). BP and FC deteriorated after the DT period (*p* < 0.05). Conclusion: Nine months of multicomponent exercise were sufficient to improve functional capacity and promote benefits in blood pressure, although was not sufficient to allow BP to reach the normal values of older women. The three month DT period without exercise caused the reversal of BP improvements but maintained the functional capacity of older women.

## 1. Introduction

The average life expectancy and the number of older individuals have increased considerably during the past two decades, reinforcing the necessity to ensure the quality of life in older people [1,2]. Regular physical activity can reduce the effects of aging and a sedentary lifestyle, and prevent the development and progression of diseases and disabling conditions [3].

Among the most significant problems, cardiovascular diseases are the leading cause of death globally, and affect more women than men. The incidence of cardiovascular diseases is higher in older women due to menopause, which accentuates the negative changes in the metabolic profile and body composition [2,3]. Cardiovascular diseases are responsible for one-third of all deaths in women [4]. The main risk factors for the development of cardiovascular disease are high blood pressure and cholesterol, overweight and obesity, diabetes mellitus, and physical inactivity [1,2]. Hypertension is present in almost 70% of older adults and, despite therapeutic medication, these older adults have a high risk of cardiovascular events and mortality [5].

Therefore, it is crucial to create and apply interventions against these cardiovascular risks. Specifically, exercise, therapeutic medication, and healthy dietary behavior are the key factors to improve health and the quality of life of older women. Several studies have shown that physical activity, via aerobic exercise, resistance training, and/or METP, can have the same effect on lowering systolic blood pressure as antihypertensive therapy [5,6,7]. Exercise also has an acute effect in lowering the blood pressure of older adults for about 22 h, known as postexercise hypotension (PEH), and can chronically increase functional capacity (FC). These actions have a clinical implication for preventing cardiovascular deleterious events [5,7,8]. METP is an exercise method that combines aerobic and resistance training, and is a low-cost intervention performed with own body weight, free weights, and elastic bands that can induce significant benefits in older women with or without hypertension [1,2,3,7].

However, because this population group does not maintain adherence to regular physical exercise programs for long periods, both obtained metabolic and functional adaptations can decline after short detraining (DT) periods [9,10,11].

Therefore, the purpose of this longitudinal study was to investigate the effects of a multicomponent training and detraining period in older women with hypertension. Our hypothesis was that older women can significantly improve their hemodynamic health profile and functional capacity with multicomponent exercise over a period of 9 months of training, and maintain it following 3 months of detraining.

## 2. Materials and Methods

### 2.1. Sample

Forty-five Caucasian women aged between 60 and 70 years, functionally independent, volunteered to participate in this study. All participants underwent a medical evaluation when attending the experimental protocol program. The exclusion criteria were: (a) have already participated in any physical activity program; (b) osteoarticular dysfunction that may interfere with the execution of the proposed motion; (c) heart problems for which the exercise prescription is not recommended; and (d) medical contraindication. The inclusion criteria were: (a) have hypertension. The study was carried out according to the Helsinki Declaration and, prior to data collection, volunteers were informed about the procedures of the study and signed an informed consent. They were advised to maintain their previous lifestyle throughout the study, including dietary patterns and physical routines. Coffee, tea, alcohol and tobacco consumption, and strenuous exercise were prohibited 24 h before the experimental procedures.

### 2.2. Procedures

The participants were separated into two groups: experimental (EG: *N* = 28; 66.7 ± 5.3 years, 1.59 ± 0.11 m), engaged in a nine month multicomponent exercise training program followed by a three month detraining period; and the control group (CG: *N* = 17; 68.2 ± 4.7 years, 1.57 ± 0.16 m), which performed no exercise (Table 1).

Data were collected by the same examiner, under the same environmental conditions (10:00–12:00 h; 22–24 °C; 55–65% humidity), prior of the multicomponent program, after nine months (at the end of the multicomponent exercise program and the beginning of the detraining period), and post three months of detraining. The same instruments were used at all times for measurements of anthropometric parameters (weight; height; body mass index), hemodynamic profile (blood pressure—systolic and diastolic pressure; resting heart rate), and functional capacity.

#### 2.2.1. Multicomponent Exercise Training Program (METP)

METP consisted of 45 min sessions undertaken twice per week over nine consecutive months, for a total of 86 sessions. The program was conducted by a specialist in physical education training for older adults, according to the ACSM guidelines for exercise prescription [2]. Each training session consisted of aerobic and muscle resistance training, with appropriate music for the activity, age, and interests of the participants, and structured as follows:(1)5–8 min of global warm-up activity, including slow walking, calisthenics, and stretching exercises.(2)15–25 min of cardiorespiratory workout (aerobic choreography with moderate intensity), with intensity maintained at 2–3 of the adapted Borg Rating of Perceived Exertion scale (RPE) in the first month, and gradually increased to 4–5 in the adapted Borg RPE.(3)15–20 min of resistance training with exercises performed in a circuit, involving exercises for the upper and lower body, agility, mobility, coordination, and social interaction, with a 20 to 30 s rest period between sets. Participants performed the weight resistance training using their own body weight (open and closed kinetic chain exercises) and elastic bands. Training intensity was progressive, particularly in the first month of training, to allow proper familiarization with the exercises and the correct and safe techniques of execution and breathing. The series and repetitions were increased each month, from 2 to 4 series and from 16 to 30 repetitions.(4)5–10 min of relaxation techniques and stretching for the upper and lower body. Static and dynamic stretching techniques were included in flexibility training.

#### 2.2.2. Detraining Period (DP)

The DP took place over three consecutive months coinciding with the summer holidays. All volunteers were instructed to conduct their normal lifestyles, including dietary patterns and physical routines, and avoid any type of systematic exercise. Participants were systematically contacted to ensure that they were fulfilling the DP requirements. The testing assessment procedures of DT were performed under the same conditions as for the METP.

#### 2.2.3. Hemodynamic Profile 

The measurements of blood pressure, systolic blood pressure (SBP), and diastolic blood pressure (DBP) were conducted using a digital sphygmomanometer (Omron Digital Blood Pressure Monitor HEM-907, Matsusaka, Japan). Resting heart rate (HR_rest_) was also measured. These measurements were taken three times under the same conditions in a seated resting position and with the left arm in support, after at least 5 min (AHA, 2005). A scale (OMRON BF 303, Matsusaka, Japan), a stadiometer (Seca, Hamburg, Germany), and bioelectrical impedance analysis were used for body mass (kg), height (cm), and body mass index (BMI, kg·m^−2^) and body fat percentage (BF, %), respectively.

#### 2.2.4. Functional Capacity Battery Test

The Functional Capacity Battery Test consisted of the Senior Fitness Test [12]. This protocol comprises six motor tests: upper limbs strength (arm curl); lower limbs strength (30 s chair stand); upper limbs flexibility (back scratch); lower limbs flexibility (chair sit-and-reach); agility/dynamic balance (8 foot up-and-go); and aerobic capacity (6 min walk).

### 2.3. Statistical Analysis

We used SPSS 19.0 for Windows (SPSS Inc., Chicago, IL, USA) for data analysis. Descriptive procedures of central tendency and dispersion were used to characterize the variable values and the normality of our sample was verified by the Shapiro–Wilk test. For inferential analysis of data, we used repeated measures ANOVA to compare, within and between groups, the mean values of each variable over the time, followed by the post-hoc Bonferroni test. The sphericity assumption was verified through the Mauchly test. The meaningfulness of the outcomes was estimated through the effect size (ES, Cohen´s d, mean divided by the standard deviation): 0.2 or less is a small ES, about 0.5 is a moderate ES, and 0.8 or more is a large ES. The delta percentage (∆%) was calculated via the standard formula: ∆% = [(posttest score – pretest score)/pretest score] × 100. For all statistical procedures, the statistical significance accepted was *p* ≤ 0.05.

## 3. Results

The participants of EG completed the exercise program with an attendance of 88%. After METP, all values of the health profiles were significantly better (*p* < 0.05) than those observed at the beginning of the study (Table 2) but not after three months of detraining (Table 2 and Table 3).

After the multicomponent exercise program, body mass index (BMI), percentage (BF%), resting heart rate (HR_rest_), and systolic (SBP) and diastolic blood pressure (DBP) improved significantly (*p* < 0.005) and declined significantly after detraining (*p* < 0.05), losing all of the benefits acquired over nine months of ME. The functional capacity (FC) improved with the program and decreased significantly after three months of detraining, but improved compared with the baseline values. The CG group maintained the baseline values throughout the study.

## 4. Discussion

The aim of this study was to analyze the changes in the hemodynamic health profile and functional capacity produced during nine month multicomponent training and three months detraining periods in older Caucasian women with hypertension. The major finding indicated that a systematic supervised multicomponent exercise program with nine months of training in older women with hypertension resulted in better blood pressure regulation, to near to normal values, and improved FC. However, three months of detraining were sufficient to return blood pressure to baseline values.

Our hemodynamic and FC improvements can be explained by the combined types of exercise of METP, which promote the specific benefits of aerobic exercise and resistance training. In the hemodynamic profile, the improvements found in HR_rest_ and blood pressure (systolic pressure: −5.37%; diastolic pressure: −5.62%) as a result of the training period are in agreement with other studies [8,13,14] that found that exercise helps to lower blood pressure. According to the meta-analysis of Kelley and Kelley [15], resistance training decreases SBP and DBP by 2% and 4%, respectively. Cornelissen et al. [7] verified decreases of 3.2 and 3.5 mmHg in SBP and DBP, respectively, and a reduction in resting heart rate due to cardiorespiratory improvements. Each meta-analysis shows that exercise can decrease SBP and DBP in normotensive and hypertensive older adults. Although the mechanisms underlying the lowering effect of exercise on SBP are not fully understood, neuro-hormonal [16,17], structural, and functional vascular adaptations were recently proposed as possible mechanisms [17]. As functional adaptations, it was suggested that aerobic training may possibly reduce sympathetic nervous activity and subsequent release of norepinephrine, lower endothelin-1 levels [18], and increase nitric oxide production [19], thereby reducing vasoconstriction and peripheral vascular resistance. According to Sosner et al. [8], aerobic exercise can reduce systolic blood pressure by 5.09 mmHg, regardless of intensity, duration and frequency of exercise, and this benefit is more substantial if a weight loss is present. Cornilissen et al. [13] reported that exercise can reduce systolic pressure by 6.9 mmHg. This reduction is higher in hypertensive individuals than in normotensive individuals [20], and can be higher in individuals having higher high blood pressure values. Furthermore, taking antihypertensive medication does not promote changes in blood pressure during the day [8]. Leitão et al. [10] reported a reduction of 3.80% in systolic pressure and 5.48% in diastolic pressure after nine months of MEP. Delgado-Floody et al. [21] reported a reduction of 7.1 mmHg in systolic pressure and 5.43 mmHg in diastolic pressure after a 16 week HIIT intervention, booth in normotensive individuals. In hypertensive subjects, Delgado-Floody et al. [21] reported a reduction of 8.70 mmHg in systolic pressure and 4.9 mmHg in diastolic pressure, similar to the results of our study. Cornelissen and Smart [7] only reported a decrease of 2.2 mmHg in systolic pressure with concurrent training. A 3 mmHg reduction can promote the reduction in stroke mortality and coronary heart disease by 8% and 5%, respectively [8], and can be maximized if exercise is applied as a long-term practice [3]. The blood pressure in our study did not reach normative values with nine months of METP, which may indicate that older women with hypertension may need a longer period of exercise to achieve these values. The FC improvements observed were similar to those of Hortobágyi et al. [22] and Marcos-Pardo et al. [23]. After 8 months of METP, Blasco-Lafarga et al. [24] reported improvements in lower body strength (20.23%), agility (22.32%), and aerobic capacity (14.54%), and Leitão et al. [10] observed a 11.06% improvement in aerobic capacity after 9 months of MEP, both in normotensive older women.

After detraining, blood pressure returned to near to baseline values in hypertensive older women. This result was not previously found in other studies as a result of the decline reported after detraining. Elliot et al. [25] reported a 9.32% increase in systolic pressure and a 1.52% increase in diastolic pressure after the suspension of a resistance training program, and Leitão et al. [10] reported a 4.13% increase in systolic pressure and a 3.38% increase in diastolic pressure after nine months of MEP. Both studies involved normotensive older women, and these results may indicate that normotensive older adults respond in a different way to detraining than hypertensive older women. In our study, FC declined with detraining but maintained a portion of the improvements observed with the MEP, results that were similar to those of other studies [9,10,26,27]. Martinez-Aldao et al. [28] reported −5.6% lower body strength, −4.6% agility, and −1.3% aerobic capacity after 8 months of MEP. Esain et al. [29], after 9 months of MEP, reported a decline of 5.8% in agility and 3.1% in upper body strength. These modifications can have important health consequences for older adults via an increased in the risk of cardiovascular disease [9,10,26,29,30,31]. These results suggest that exercise focused on MEP can produce a strong positive effect by protecting older women with hypertension against health declines associated with age, and that detraining periods should be avoided [10].

## 5. Conclusions

The hemodynamic health profile and functional capacity of older women with hypertension can be improved, to those of normotensive older women, through nine months of a multicomponent exercise program. However, such a program was not sufficient to achieve normal BP values, and a three month detraining period was sufficient to eliminate the blood pressure benefits in hypertensive older women. METP is a low-cost and effective exercise method that can be applied in the community. Interruptions should be avoided when prescribing exercise to older adults to maintain or reduce the negative impact of detraining and their risk of heart disease, and to maintain or increase their daily activities, health, and quality of life.

## Figures and Tables

**Table 1 ijerph-18-09117-t001:** Subject’s anthropometric characteristics.

Variable	Group	Baseline (BL)	Post Exercise (PE)	Post Detraining (PD)
Body weight (kg)	EG	72.60 ± 9.12	71.49 ± 9.19	71.95 ± 9.22
	CG	71.58 ± 10.29	71.44 ± 10.55	71.97 ± 10.50
BMI (kg·m^−2^)	EG	30.67 ± 3.18	30.20 ± 4.19	30.39 ± 5.50
	CG	29,76 ± 5.00	29,70 ± 5.13	29.91 ± 5.10
BF (%)	EG	39.12 ± 2.42	37.69 ± 2.17	38.29 ± 2.14
	CG	39.01 ± 2.14	38.95 ± 2.04	39.26 ± 1.99

Data presented are mean ± SD; body mass index (BMI); body fat percentage (BF).

**Table 2 ijerph-18-09117-t002:** Delta percentage differences of parameters after MEP and detraining period.

	Group	SBP (%)	DBP (%)	HR_rest_ (%)	UBS (%)	LBS (%)	UBF (%)	LBF (%)	2TUG (%)	6MWT (%)
BL vs. PE	EG	−5.4 *	−5.6 *	−7.7 *	10 *	27.8 *	53.9 *	100 *	−9.8 *	8.6 *
	CG	−0.4	−0.1	0.3	1.1	−5.9	7	17.1	−0.7	−0.7
PE vs. PD	EG	1.1 *	0.1 *	7 *	−4.5 *	−13 *	−33.3 *	−37.5 *	3.5 *	−4.8 *
	CG	0.4	0.2	0.6	0.4	1.1	−6.4	−14.2	−0.5	−0.3

Data presented are the delta percentages (∆%): (a) before the nine month multicomponent exercise program (BL) and post exercise/beginning of detraining (PE); (b) post exercise and post detraining (PD) of systolic blood pressure (SBP), diastolic blood pressure (DBP), resting heart rate (HR_rest_), upper body strength (UBS), lower body strength (LBS), upper body flexibility (UBF), lower body flexibility (LBF), agility/dynamic balance (2TUG), aerobic endurance six-minute walk test (6MWT); * *p* < 0.05.

**Table 3 ijerph-18-09117-t003:** Effects of the multicomponent exercise program and detraining in the hemodynamic health profile and functional capacity of older women with hypertension.

	CG							EG						
	BL	PE	PD	BL vs. PD				BL	PE	PD	BL vs. PD			
				Confidence Interval		ES	*p*				Confidence Interval		ES	*p*
				Lower	Upper						Lower	Upper		
SBP (mmHg)	144.47 ± 3.41	143.94 ± 2.36	144.12 ± 2.24	−1.59	2.54	0.12	0.64	148.96 ± 7.18	141.36 ± 3.49 *	151.86 ± 9.27 ^+^	−4.97	−0.82	0.35	0.01
DBP (mmHg)	86.47 ± 2.47	86.32 ± 2.78	86.65 ± 2.32	−0.91	0.56	0.08	0.62	88.36 ± 5.03	82.96 ± 5.00 *	84.50 ± 9.81 ^+^	0.8	6.91	0.50	0.02
HR_rest_ (bpm)	81.47 ± 3.95	81.52 ± 4.7	81.11 ± 4.06	−0.05	0.76	0.09	0.08	77.32 ± 7.38	70.96 ± 8.81 *	75.61 ± 7.81 ^+^	0.54	2.89	0.23	0.01
LBS (repetitions)	17.24 ± 2.73	16.35 ± 3.05	17.06 ± 2.35	−0.46	0.81	0.07	0.57	18.11 ± 2.36	22.79 ± 3.01 *	20.39 ± 2.61 ^+^	−2.46	−2.11	0.91	0.00
UBS (repetitions)	19.36 ± 1.27	20.35 ± 1.22	20.12 ± 2.32	−2.19	0.66	0.41	0.27	19.54 ± 1.88	21,54 ± 2.69 *	20.71 ± 2.11 ^+^	−1.97	−0.39	0.59	0.01
2TUG (s)	5.75 ± 0.20	5.71 ± 0.24	5.68 ± 0.21	−0.06	0.19	0.34	0.28	5.70 ± 0.42	5.21 ± 0.53 *	5.40 ± 0.56 ^+^	0.12	0.48	0.61	0.00
LBF (cm)	1.76 ± 2.61	2.06 ± 2.72	2.01 ± 2.55	−0.75	0.63	0.10	0.86	1.54 ± 2.17	3.64 ± 1.83 *	2.46 ± 1.75 ^+^	−2.33	−0.31	0.47	0.01
UBF (cm)	−5.88 ± 3.71	−5.47 ± 3.29	−5.82 ± 3.53	−1.17	0.70	0.02	0.60	−4.96 ± 4.28	−2.46 ± 4.74 *	−3.64 ± 3.88 ^+^	−1.52	−0.33	0.32	0.00
6MWT (m)	576.64 ± 52.89	572.35 ± 41.31	570.59 ± 47.07	−9.33	13.45	0.12	0.71	563.93 ± 56.69	623.93 ± 60.31 *	586.96 ± 59.99 ^+^	−30.26	−15.81	0.39	0.00

Data presented are mean ± SD; before multicomponent training (BL), post exercise (PE), and post detraining (PD) of systolic blood pressure (SBP), diastolic blood pressure (DBP), resting heart rate (HR_rest_), upper body strength (UBS), lower body strength (LBS), upper body flexibility (UBF), lower body flexibility (LBF), agility/dynamic balance (2TUG), aerobic endurance six-minute walk test (6MWT). * *p* < 0.05, significant improvements after training period of multicomponent training program; ^+^
*p* < 0.05, significant decreases with detraining period after multicomponent training program.

## Data Availability

The data presented in this study are available on request from the corresponding author.

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
