# Peer review of "Can Exercise Help Regulate Blood Pressure and Improve Functional Capacity of Older Women with Hypertension against the Deleterious Effects of Physical Inactivity?"

_ijerph, 2021, doi:10.3390/ijerph18179117_

Round 1

Reviewer 1 Report

To the Authors

The study of Leitao et al (1301651) investigated the effects of a multicomponent training and detraining period in older women with hypertension. The authors hypothesized that older women can significantly improve their hemodynamic health profile and functional capacity with multicomponent exercise over 9 months of training and keep it even following 3 months of detraining. The authors reported that regular exercise training lasting 9 months of training resulted in better blood pressure regulation and functional capacity was improved, while three months of detraining were enough to roll back blood pressure to baseline values. The study is well designed and nicely executed. There are, however, few data interpretational issues that authors need to clarify and subsequently address in their manuscript.

Major Points

  • The rationale of this study is not very clear.
  • The practical considerations are not so highlighted.
  • The use of the English language is occasionally inappropriate.
  • The description of the methods is not so clear.

Specific Points

The following points need to be addressed:

Abstract

  • Paragraph 1, line 21: ‘Multicomponent exercise (ME)’ What did you mean? Did you mean interval or circuit exercise? Please use a more appropriate term.
  • Paragraph 1, line 23: ‘(EG: 66.7±5.3 years, 59±0.11 cm)’ Please define first the abbreviation ‘EG’ and then use it throughout the manuscript. Furthermore, 1.59±0.11 cm is stature height? If yes,
  • please correct the unit is in meters and not cm.
  • Paragraph 1, line 25: ‘CG’ Please define first the abbreviation ‘EG’ and then use it throughout the manuscript.
  • Paragraph 1, lines 27-30: ‘ME program improved (p<0.05) Systolic BP (-5.37%), Diastolic BP(-5.67%), rest heart rate(-7.8%), agility (9.8%), lower body strength (27.8%), upper body strength (10.0%), cardiorespiratory capacity(8.6%). BP and FC deteriorated after DT period (p<0.05).’ Please be consistent and put a space between word and paraphrenitis.
  • You have mentioned that women in the control group had hypertension. What about the participants in the exercise group? All participants had hypertension?

Introduction:

The rationale of this study is not very clear. Authors have to explain why you choose women older individuals compared to males. To highlight the benefit of attending a regular exercise training program. To define the multicomponent exercise training program and which are the advantage compared to the traditional exercise training program. Furthermore, to highlight the practical considerations of this research.

  • Paragraph 1, lines 40-41: ‘regular physical activity can reduce aging effects related to aging’, Please rewrite it.
  • Paragraph 2, lines 43-44: The differences between males and females in cardiovascular disease and thus in morbidity and mortality demonstrated mostly after menopause. Please, be more specific about it.
  • Paragraph 3, lines 50-52: ‘Therefore, it is essential to create and apply interventions against these cardiovascular risk incidences, being exercise, therapeutic medication and healthy dietary behavior the key factors.’ Please rewrite it, is not so clear the meaning of the sentence.
  • Paragraph 3, line 52: What kind of exercise? Aerobic, resistance, high-intensity, or moderate-intensity exercise? Please specific it.

Materials and Methods.

Sample

  • Paragraph 1, lines 71-72: ‘heart problems where the exercise prescription injures the health of the older’ Does not make sense, please rewrite it.

The aim of this study was to investigate the effect of the exercise training program in individuals with hypertension. The authors should provide more information about the baseline arterial blood pressure of the participants. Which are the inclusion criteria of this study? What about the comorbidity due to high arterial blood pressure? What about the medication of the participants due to hypertension? Did participants had diagnosed with high blood pressure and were on optimal medication before participation in this study?

Procedures

  • Paragraph 1, lines 80 and 82: ‘1.59±0.11 cm and 1.57±0.16 cm’ Please specify which variable is this? Is stature height? If yes, correct the unit is meter, not cm.
  • Paragraph 2, line 87: ‘eight’ I think that something is missing.
  • Table 1: Please use baseline instead of before exercise, post exercise instead of beginning detraining, post detraining instead of after detraining throughout the manuscript.
  • Table 1: The appropriate unit for stature height (1.57 and 1.59) is m and not cm.
  • Table 1 footnotes: ‘confidence interval of the difference between BD and AD, Δ% (value of the difference between BD and AD), Effect Size (ES), p-value of detraining effect; before detraining (BD) and after detraining (AD) of body fat percentage (BF%) body mass index (BMI) and body weight (kg)’ All these information did not show in the Table. Table 1 represents only the mean values ±SD.

Multicomponent exercise training program (METP)

  • Lines 109-110: ‘15-20 minutes of resistance training (exercises for the upper and lower body) with exercises performed in a circuit, involving exercises for the upper and lower body’ Please rewrite it, authors give the same information twice.
  • Did the participants perform any evaluation at the midtime of the exercise training program in order to adjust the training intensity and to see the adaptation?
  • All participants finished the study or where was a dropout?
  • What was the attendance rate for training groups? Please provide the information.

Detraining Period (DP)

  • Lines 124-125: ‘The testing assessment procedures after DP were collected at the same conditions as for the METP’ Please rewrite it.

Hemodynamic Profile

  • Which are the interclass correlation coefficient (ICC) for test-retest reliability for blood pressure, anthropometrics, and body composition evaluation?
  • Body composition was assessed through a bioimpedance device. Please provide information about whether all participants followed the same instruction before, after the exercise training program and after detraining for accurate body composition measurements.

Functional Capacity Battery Test

Authors should provide more information about the procedures of exercise testing.

Statistical Analysis

  • Lines 143-146: ‘For inferential analysis of data, we used repeated measures ANOVA to compare the mean values of each variable in each year of study and to compare between each of the three years of study, followed by the post-hoc Bonferroni test.’ Authors should mention the total duration of this study. How did three years of research come about?

Results

  • Line 154: ‘This All the participants completed the MEP with an attendance of 88%’ Please rewrite the whole sentence.
  • Line 154: ‘multicomponente’ Please correct it, multicomponent.
  • Table 2. The magnitude of changes in physiological variables are different between the exercise and control group? Authors should provide this information both in tables and main text.
  • Line 164: ‘Body Fat index (BMI)’ Please correct it, is body mass index.
  • Table 3: It is not able to read it. Author should make it again. Provide more information on whether there are differences between the exercise and control group.

Discussion

It is well written without raising major concerns.

Author Response

We are grateful for your consideration of this manuscript, and we also very much appreciate your suggestions, which have been very helpful in improving the manuscript. We also thank the reviewers for their careful reading of our text. All the comments we received on this study of all reviewers have been attended into account in improving the quality of the article, and we present our reply to each of them separately.

Abstract - DONE

  • Paragraph 1, line 21: ‘Multicomponent exercise (ME)’ What did you mean? Did you mean interval or circuit exercise? Please use a more appropriate term.

DONE – We changed to METP

  • Paragraph 1, line 23: ‘(EG: 66.7±5.3 years, 59±0.11 cm)’ Please define first the abbreviation ‘EG’ and then use it throughout the manuscript. Furthermore, 1.59±0.11 cm is stature height? If yes, please correct the unit is in meters and not cm.

DONE – We changed.

  • Paragraph 1, lines 27-30: ‘ME program improved (p<0.05) Systolic BP (-5.37%), Diastolic BP(-5.67%), rest heart rate(-7.8%), agility (9.8%), lower body strength (27.8%), upper body strength (10.0%), cardiorespiratory capacity(8.6%). BP and FC deteriorated after DT period (p<0.05).’ Please be consistent and put a space between word and paraphrenitis.

DONE

  • You have mentioned that women in the control group had hypertension. What about the participants in the exercise group? All participants had hypertension?
  • DONE – We add Hypertension in EG

Introduction:

The rationale of this study is not very clear. Authors have to explain why you choose women older individuals compared to males. To highlight the benefit of attending a regular exercise training program. To define the multicomponent exercise training program and which are the advantage compared to the traditional exercise training program. Furthermore, to highlight the practical considerations of this research.

DONE: Line 43-46; Line 51-72

  • Paragraph 1, lines 40-41: ‘regular physical activity can reduce aging effects related to aging’, Please rewrite it.

DONE: We rewrite.

  • Paragraph 2, lines 43-44: The differences between males and females in cardiovascular disease and thus in morbidity and mortality demonstrated mostly after menopause. Please, be more specific about it.

DONE: We add paragraph 44-45

  • Paragraph 3, lines 50-52: ‘Therefore, it is essential to create and apply interventions against these cardiovascular risk incidences, being exercise, therapeutic medication and healthy dietary behavior the key factors.’ Please rewrite it, is not so clear the meaning of the sentence.

DONE: line 51-54

  • Paragraph 3, line 52: What kind of exercise? Aerobic, resistance, high-intensity, or moderate-intensity exercise? Please specific it.

Done: line 54

Materials and Methods.

Sample

  • Paragraph 1, lines 71-72: ‘heart problems where the exercise prescription injures the health of the older’ Does not make sense, please rewrite it.

DONE: line 87

The aim of this study was to investigate the effect of the exercise training program in individuals with hypertension. The authors should provide more information about the baseline arterial blood pressure of the participants. Which are the inclusion criteria of this study? What about the comorbidity due to high arterial blood pressure? What about the medication of the participants due to hypertension? Did participants had diagnosed with high blood pressure and were on optimal medication before participation in this study?

DONE: Line 88. We add inclusion criteria. All participants underwent a medical evaluation before the METP.

Procedures

  • Paragraph 1, lines 80 and 82: ‘1.59±0.11 cm and 1.57±0.16 cm’ Please specify which variable is this? Is stature height? If yes, correct the unit is meter, not cm.

DONE: We changed

  • Paragraph 2, line 87: ‘eight’ I think that something is missing.

DONE: we changed to Height

  • Table 1: Please use baseline instead of before exercise, post exercise instead of beginning detraining, post detraining instead of after detraining throughout the manuscript.

DONE: we changed

  • Table 1 footnotes: ‘confidence interval of the difference between BD and AD, Δ% (value of the difference between BD and AD), Effect Size (ES), p-value of detraining effect; before detraining (BD) and after detraining (AD) of body fat percentage (BF%) body mass index (BMI) and body weight (kg)’ All these information did not show in the Table. Table 1 represents only the mean values ±SD.

DONE: we changed

Multicomponent exercise training program (METP)

  • Lines 109-110: ‘15-20 minutes of resistance training (exercises for the upper and lower body) with exercises performed in a circuit, involving exercises for the upper and lower body’ Please rewrite it, authors give the same information twice.

DONE: WE CHANGED

  • Did the participants perform any evaluation at the midtime of the exercise training program in order to adjust the training intensity and to see the adaptation?

DONE: No, we only evaluate in the beginning and at the end of the exercise program.

  • All participants finished the study or where was a dropout?

DONE: All older women with hypertension finished the study.

  • What was the attendance rate for training groups? Please provide the information.

DONE: The attendance rate was 88% - Line 199

Detraining Period (DP)

  • Lines 124-125: ‘The testing assessment procedures after DP were collected at the same conditions as for the METP’ Please rewrite it.

DONE: Line 148. We rewrited.

Hemodynamic Profile

  • Which are the interclass correlation coefficient (ICC) for test-retest reliability for blood pressure, anthropometrics, and body composition evaluation?
  • Body composition was assessed through a bioimpedance device. Please provide information about whether all participants followed the same instruction before, after the exercise training program and after detraining for accurate body composition measurements.

DONE: Line 154.

Statistical Analysis

  • Lines 143-146: ‘For inferential analysis of data, we used repeated measures ANOVA to compare the mean values of each variable in each year of study and to compare between each of the three years of study, followed by the post-hoc Bonferroni test.’ Authors should mention the total duration of this study. How did three years of research come about?

DONE: We rewrited. Line-188-199

Results

  • Line 154: ‘This All the participants completed the MEP with an attendance of 88%’ Please rewrite the whole sentence.

DONE: we rewrite

  • Line 154: ‘multicomponente’ Please correct it, multicomponent.

DONE

  • Table 2. The magnitude of changes in physiological variables are different between the exercise and control group? Authors should provide this information both in tables and main text.

DONE: This information is in table 3.

  • Line 164: ‘Body Fat index (BMI)’ Please correct it, is body mass index.

DONE: we rewrited.

  • Table 3: It is not able to read it. Author should make it again. Provide more information on whether there are differences between the exercise and control group.

DONE: There were no differences in baseline values between EG and CG.

Reviewer 2 Report

Major: While authors reported that long-term exercise effectively lowered blood pressure in older women with hypertension, many studies have reported similar results as the authors have mentioned in discussion section. What else is new about this study? What is the advantage of multi component exercise program over conventional exercise program? The authors should discuss the significance and strength of the present study rather than merely reviewing what is always known. The discussion section should be more focused on the authors’ own work. Minor: 1. Table 2. The absolute value of their difference should also be provided. 2. Table3 is too busy to read. Please correct it. What does round to the second decimal place mean?

Author Response

We are grateful for your consideration of this manuscript, and we also very much appreciate your suggestions, which have been very helpful in improving the manuscript. We also thank the reviewers for their careful reading of our text. All the comments we received on this study of all reviewers have been attended into account in improving the quality of the article, and we present our reply to each of them separately.

While authors reported that long-term exercise effectively lowered blood pressure in older women with hypertension, many studies have reported similar results as the authors have mentioned in discussion section. What else is new about this study?

​The novelty of our study is the fact that volunteers were hypertensive and women. Most part of the studies have investigated only older subjects. 

What is the advantage of multi component exercise program over conventional exercise program?.

Due to a chronic training program, including cardiovascular and resistance exercises, the possible physiological effects are better than only one type of training. Also, it should highlight the low cost and feasibility of this type of intervention. 

The authors should discuss the significance and strength of the present study rather than merely reviewing what is always known. The discussion section should be more focused on the authors’ own work

Our study can bring new knowledge on how hypertensive older women (In the majority of the studies the participants aren´t hypertensive so we think this is a major point of our study) can benefit from exercise specifically from METP, and how detraining affects this specific age group. The benefits of using METP may be caused from the combination of the benefits that results from aerobic exercise and resistance training. We observed that older women with hypertension may need more months of METP to attain normative blood pressure and that three months of detraining reversed all blood pressure benefits (in normotensive older adults this may not occur). To support this we add paragraph 69-72; 312-314; 343-345; 367-372.

Round 2

Reviewer 1 Report

The authors have provided a thorough revision of the manuscript which has been substantially improved. The author’s response to all major and minor points. However, they should pay attention to the above.

Abstract

  • Paragraph 1, lines 24, 28: The correct abbreviation is METP instead of ME. Please correct it.
  • Did not use abbreviation in abstract without first define it (i.e., EG, CG), otherwise delete it.

Introduction:

  • Paragraph 2, lines 43-46: ‘Among main problems, cardiovascular diseases….. negative changes in metabolic profile and body composition’ Please, provide a reference.

Materials and Methods.

Sample

Did participants had diagnosed with high blood pressure and were on optimal medication before participation in this study?

Procedures

  • Paragraph 2, line 92: (PE). Please delete it.
  • Paragraph 2, line 95: BMI, Please define it.
  • Table 1: Please define the abbreviation of Table 1 in footnotes.

Multicomponent exercise training program (METP)

  • Line 124: ‘for agility’ Please delete the for.

Hemodynamic Profile

  • Line 149: ‘(BF, %) instead of (%BF, %). Please correct it.
  • Which are the interclass correlation coefficient (ICC) for test-retest reliability for blood pressure, anthropometrics, and body composition evaluation?

Results

  • Line 192: body mass index instead of Body mass index. Please correct it.
  • Lines 192-193: ‘and percentage (BF%)’ Something is missing, does not make sense. Please correct it.
  • Table 2. Put the significant signal (*) between the main groups if there are significant differences between the groups as you state in the main text.
  • Line 197. maintained the baseline values instead of maintained their values. Please correct it.
  • Table 3: BL vs PD instead of BE vs AD. Please correct it.

Discussion

  • Line 253: which instead of witch. Please correct it.
  • Line 256: METP instead of MEP. Please correct it.

Author Response

We are grateful for your consideration of this manuscript, and we also very much appreciate your last suggestions, which have been attended and were very helpful in improving the manuscript. We also thank the reviewers for their careful reading of our text. 

Abstract

We Changed

Introduction

We Changed

Materials and Methods

We Changed. All participants were approved by the medical staff to attend the program.

Results

We Changed.

Discussion

We Changed.

Reviewer 2 Report

Thank you for responding my comments. The authors have addressed most of my concerns, but still I have one concern: What does the second decimal place in the numbers in the tables mean?  If that is not essential, round off the numbers to one decimal place. The tables are still busy.

Author Response

We are grateful for your consideration of this manuscript, and we also very much appreciate your last suggestions, which have been attended in part and were very helpful in improving the manuscript. We also thank the reviewers for their careful reading of our text.

You refer that the tables are busy but we need to mantain all the data.  In table 2 we round the percentage numbers to one decimal place and in the other tables we decided to mantain the two decimal place.